# A Mean-Field Variational Inference Approach to Deep Image Prior for Inverse Problems in Medical Imaging

**Malte Tölle**[*1,2]             MALTE.TOELLE@MED.UNI-HEIDELBERG.DE

**Max-Heinrich Laves**[*3]            MAX-HEINRICH.LAVES@TUHH.DE

**Alexander Schlaefer**[3]            SCHLAEFER@TUHH.DE

[1] *Department of Internal Medicine III, Heidelberg University Hospital*

[2] *Informatics for Life, Heidelberg*

[3] *Institute of Medical Technology and Intelligent Systems, Hamburg University of Technology*

**Editors:** Under Review for MIDL 2021

## Abstract

Exploiting the deep image prior property of convolutional auto-encoder networks is especially interesting for medical image processing as it avoids hallucinations by omitting supervised learning. Its spectral bias towards lower frequencies makes it suitable for inverse image problems such as denoising and super-resolution, but manual early stopping has to be applied to act as a low-pass filter. In this paper, we present a novel Bayesian approach to deep image prior using mean-field variational inference. This allows for uncertainty quantification on a per-pixel level and, given the right prior distribution on the network weights, omits the need for early stopping. We optimize the parameters of the weight prior towards reconstruction accuracy using Bayesian optimization with Gaussian Process regression. We evaluate our approach on different inverse tasks on a variety of modalities and demonstrate that an optimized weight prior outperforms former state-of-the-art Bayesian deep image prior approaches. We show that a badly selected prior leads to worse accuracy and calibration and that it is sufficient to optimize the weight prior parameter per task domain.

**Keywords:** Variational inference, Hallucination, Deep learning

## 1. Introduction

Automated methods for improving image quality have several applications in medical imaging, as acquiring high-quality images is time-consuming, costly, or entails a considerable radiation dose to the patient. Such use cases include denoising and artifact removal in low-dose CT or PET (Yang et al., 2018; Ma et al., 2020; Wang et al., 2018), despeckling in ultrasound or optical coherence tomography (Michailovich and Tannenbaum, 2006; Bernardes et al., 2010), super-resolution of MRI (Tanno et al., 2017), or inpainting for hair removal in dermoscopy images (Abbas et al., 2011). Enhancing medical images with poor quality is a fundamental step for better diagnosis or subsequent image analysis. In this paper, we focus on post-processing methods that are generally applicable to all aforementioned modalities.

Those methods involve solving an inverse imaging problem, which try to reconstruct a high-quality image $\hat{\boldsymbol{x}}$ from a low-quality observation $\tilde{\boldsymbol{x}} = \boldsymbol{c} \circ \boldsymbol{x}$ of the true, but unknown

---

[*] Contributed equally

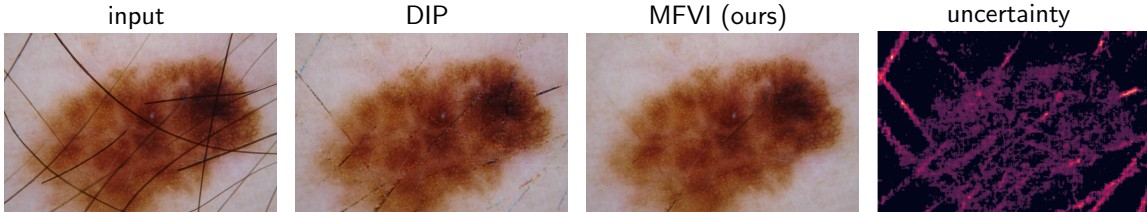

input      DIP      MFVI (ours)      uncertainty

Figure 1: Inpainting for hair removal on dermoscopy images. Our mean-field variational inference approach to deep image prior is not prone to overfitting, outperforms the non-Bayesian baseline and provides consistent pixel-wise uncertainty maps.

image $\boldsymbol{x}$ affected by some corruption process $\boldsymbol{c}$. The reconstruction comprises minimization of an objective function $\hat{\boldsymbol{x}} = \arg\min \mathcal{L}(\tilde{\boldsymbol{x}}, \hat{\boldsymbol{x}}) + \lambda \mathcal{R}(\hat{\boldsymbol{x}})$, governed by a similarity measure $\mathcal{L}$ and some regularizing image prior $\mathcal{R}$, weighted by a factor $\lambda$ (Sotiras et al., 2013). Common priors for image quality enhancement are total variation or penalization of first and higher order spatial derivatives (Rudin et al., 1992). The prior is of particular importance as it is responsible for the properties of the enhanced image; its manual selection is a delicate task.

More recently, deep-learning-based convolutional autoencoders have been trained to enhance images using sets of corrupted and uncorrupted data pairs (Jain and Seung, 2009). Autoencoders extract important visual features from the corrupted input image and reconstruct the input from the extracted features using learned image statistics. Through this, the neural networks implicitly learn regularization priors from data.

However, deep-learning-based methods show insufficient robustness to input data that lay outside their training domain. Antun et al. (2020) have demonstrated that state-of-the-art deep learning methods for CT and MR image reconstruction, such as AUTOMAP (Zhu et al., 2018), show severe instabilities to tiny perturbations in the input data, which causes the reconstructions to contain considerable artifacts. Even worse, novel pathologies that were not present in the training data can be made to disappear in the reconstruction (Bhadra et al., 2020). This phenomenon is referred to as *hallucination* and is not limited to tomographic reconstruction but also happens in other deep-learning-based inverse image tasks (Laves et al., 2020b). Hallucinations can result in misdiagnosis and must be avoided at all costs in medical imaging.

## 1.1. Related Work

Lempitsky et al. (2018) have shown that the excellent performance of deep convolutional networks for inverse image tasks on in-domain data is not only due to their ability to learn image priors from data, but also due to the structure of the networks themselves. The concept of deep image prior (DIP) for inverse tasks does not require supervised training and thus, it is not affected by the aforementioned instabilities and hallucinations. Besides empirical evidence, the effectiveness of DIP can be explained by the spectral bias of deep networks (Rahaman et al., 2019). An autoencoder network decouples the frequency components of an image, comparable to a Fourier transform (Chakrabarty and Maji, 2019). During optimization, the frequency components are learned at different rates. Lower frequencies are

reconstructed first, which behaves like a low pass filter; image corruptions such as noise are usually encoded in the high-frequency components. This makes early stopping in optimization a crucial step in order to not overfit the corrupting features (see Fig. 1).

However, early stopping requires expert human interaction. We seek to find a more automated way to prevent DIP from overfitting in order to take advantage of its robustness towards hallucinations. Cheng et al. (2019) presented a first Bayesian approach to DIP in the context of natural images, where a prior distribution is placed over the weights of the network and the posterior distribution is used to output the final image. They derived a Monte Carlo (MC) sampler from DIP using stochastic gradient Langevin dynamics (SGLD) as Bayesian approximation, which uses injection of Gaussian noise into the gradients during each SGD step (Welling and Teh, 2011). The authors claim to have solved the problem of overfitting and provide pixel-wise reconstruction uncertainty estimates. SGLD DIP has already been applied to PET image reconstruction (Carrillo et al., 2021). Recently, Laves et al. (2020b) have shown that DIP with SGLD shows almost unchanged overfitting behavior in the case of medical images. As a solution, they proposed a variational inference (VI) approach to DIP using Monte Carlo dropout (Gal and Ghahramani, 2016).

In this paper, we show that former Bayesian approaches to DIP show overfitting on medical images at some point. We attribute this to the manual selection of the weight's prior distribution. It is important to distinguish between DIP, which imposes a spectral bias towards lower frequencies, and the prior distribution over the weights of the network in Bayesian inference. In SGLD and MC dropout, the prior is implicitly defined by weight decay or the dropout rate. We hypothesize that the potential of DIP can be utilized in medical image enhancement using a well-defined prior distribution in a Bayesian setting.

**Contributions** Our contribution is a novel approximate Bayesian approach to DIP by employing mean-field VI (MFVI), where the weight prior can be defined more explicitly than in SGLD or MC dropout. We further use Bayesian optimization (BO) to tune the parameters of the weight prior on a per-task level and show its superiority to former approaches on different medical image enhancement problems. Our code is available at github.com/maltetoelle/mfvi-dip.

### 1.2. Background

**Bayesian Deep Learning** See Appendix F for background information about Bayesian deep learning.

**Deep Image Prior** Convolutional networks have been extensively used to learn image priors from data. Lempitsky et al. (2018) have shown that the structure of a CNN is sufficient to capture a great amount of image statistics and impose a strong prior to restore a high-quality image from a low-quality observation without having access to any data. An image-generating network $\hat{\boldsymbol{x}} = \boldsymbol{f_w}(\boldsymbol{z})$ with randomly-initialized weights $\boldsymbol{w}$ is used as a parameterization of the image. The input $\boldsymbol{z}$ is sampled from a uniform distribution $\boldsymbol{z} \in \mathbb{R}^{C \times H \times W} \sim \mathcal{U}(0, 0.1)$ with channels $C$, width $W$ and height $H$. Given a low-quality target image $\tilde{\boldsymbol{x}}$, the reconstructed image is obtained by minimizing the pixel-wise mean squared error $\|\tilde{\boldsymbol{x}} - \boldsymbol{f_w}(\boldsymbol{z})\|^2$ w.r.t. the weights $\boldsymbol{w}$. Due to the spectral bias of DIP towards lower frequencies, early stopping behaves like a low-pass filter (Chakrabarty and Maji, 2019), making it suitable for many inverse image tasks.

## 2. Methods

### 2.1. Mean-Field Variational Inference for Deep Image Prior

Given a low-quality medical image $\tilde{\boldsymbol{x}}$ and an image-generator network $\boldsymbol{f_w}(\boldsymbol{z}) = \hat{\boldsymbol{x}}$ with randomly-initialized weights $\boldsymbol{w}$, DIP aims at finding the optimal weight point estimate $\hat{\boldsymbol{w}}$ by maximum likelihood estimation (MLE) with gradient descent. The input $\boldsymbol{z}$ has the same spatial dimensions as $\hat{\boldsymbol{x}}$. Before we turn to a Bayesian approach, we model heteroscedastic reconstruction uncertainty by assuming that $\tilde{\boldsymbol{x}}$ is sampled from a spatial random process and that each pixel $i$ follows a Gaussian distribution $\mathcal{N}(\tilde{x}_i; \hat{x}_i, \hat{\sigma}_i^2)$ with mean $\hat{x}_i$ and variance $\hat{\sigma}_i^2$. We extend the last layer such that the network outputs these values for each pixel $\boldsymbol{f_w}(\boldsymbol{z}) = \left[\hat{\boldsymbol{x}}, \hat{\boldsymbol{\sigma}}^2\right]$. Maximum posterior is performed by minimizing the negative log-likelihood, which leads to the following optimization criterion (Kendall and Gal, 2017)

$$\mathcal{L}(\boldsymbol{w}) = \frac{1}{N} \sum_{i=1}^{N} \hat{\sigma}_i^{-2} \big\|\tilde{x}_i - \hat{x}_i\big\|^2 + \log \hat{\sigma}_i^2 \ , \tag{1}$$

where $N$ is the number of pixels per image. For numerical stability, Eq. (1) is implemented such that the network directly outputs $-\log \hat{\boldsymbol{\sigma}}^2$.

Next, we employ a MFVI approach to DIP by assuming that the variational posterior can be factorized as $q_{\boldsymbol{\phi}}(\boldsymbol{w}) = \prod_{i=1}^{L} \mathcal{N}(w_i \,|\, \mu_i, \sigma_i^2)$, with number of layers $L$. In each forward pass, the weights are sampled using reparameterization $\boldsymbol{w} = \boldsymbol{\mu} + \boldsymbol{\sigma} \odot \boldsymbol{\epsilon}$ with $\boldsymbol{\epsilon} \sim \mathcal{N}(\mathbf{0}, \boldsymbol{I})$, where $\odot$ denoting element-wise multiplication. The variational parameters $\boldsymbol{\phi} = \{\boldsymbol{\mu}, \boldsymbol{\sigma}\}$ are optimized by minimizing the negative log evidence lower bound (ELBO)

$$\boldsymbol{\phi}^* = \arg\min_{\boldsymbol{\phi}} \ \mathrm{KL}[q_{\boldsymbol{\phi}}(\boldsymbol{w}) \,\|\, p(\boldsymbol{w})] - \mathbb{E}_{\boldsymbol{w} \sim q_{\boldsymbol{\phi}}}[\log p(\mathcal{D} \,|\, \boldsymbol{w})] \tag{2}$$

using backpropagation without weight decay. This effectively doubles the number of trainable parameters and is known as Bayes by backprop (Blundell et al., 2015). The first term in Eq. (2) is usually approximated with MC integration by

$$\mathrm{KL}[q\|p] \approx \frac{1}{T} \sum_{i=1}^{T} \log q_{\boldsymbol{\phi}}(\boldsymbol{w}_i) - \log p(\boldsymbol{w}_i) \ , \tag{3}$$

with $T$ Monte Carlo samples $\boldsymbol{w}_i$ drawn from the variational posterior $q_{\boldsymbol{\phi}}(\boldsymbol{w})$. In case of a Gaussian prior, it can be implemented in closed form accelerating training by omitting the need for drawing MC samples (cf. Appendix A). The second term in Eq. (2), the log likelihood, is implemented using Eq. (1) in the same MC fashion:

$$-\mathbb{E}_{\boldsymbol{w} \sim q_{\boldsymbol{\phi}}}[\log p(\mathcal{D} \,|\, \boldsymbol{w})] \approx \frac{1}{T} \sum_{i=1}^{T} \boldsymbol{\sigma}_{\boldsymbol{w}_i}^{-2} \|\tilde{\boldsymbol{x}} - \hat{\boldsymbol{x}}_{\boldsymbol{w}_i}\|^2 + \log \boldsymbol{\sigma}_{\boldsymbol{w}_i}^2 \ . \tag{4}$$

After convergence, we obtain the high-quality image $\mathbb{E}[\hat{\boldsymbol{x}}]$ and the accompanying pixel-wise uncertainty $\mathrm{Var}[\hat{\boldsymbol{x}}]$ by MC sampling from the predictive posterior (Kendall and Gal, 2017):

$$\mathbb{E}_{\boldsymbol{w} \sim q_{\boldsymbol{\phi}}}[\hat{\boldsymbol{x}}] \approx \frac{1}{T} \sum_{i=1}^{T} \hat{\boldsymbol{x}}_{\boldsymbol{w}_i} \ , \quad \mathrm{Var}_{\boldsymbol{w} \sim q_{\boldsymbol{\phi}}}[\hat{\boldsymbol{x}}] \approx \frac{1}{T} \sum_{i=1}^{T} \left( \hat{\boldsymbol{x}}_i - \frac{1}{T} \sum_{i=1}^{T} \hat{\boldsymbol{x}}_i \right)^2 + \frac{1}{T} \sum_{i=1}^{T} \hat{\sigma}_i^2 \ . \tag{5}$$

## 2.2. Prior Selection with Bayesian Optimization

Instead of manually selecting the prior distribution over the weights of the DIP network using heuristics or inefficient grid search, we employ derivative-free BO. BO allows us to optimize black-box functions that are expensive to evaluate, such as the training of a deep network (Snoek et al., 2015). It uses a computationally inexpensive surrogate model to retrieve a distribution over functions. In this work, we maximize the peak signal-to-noise ratio (PSNR) between the reconstruction $\hat{x}$ and the high-quality image $x$ as a function of the prior standard deviation $\sigma_p$

$$\max_{\sigma_p \in A} f(\sigma_p) = \max_{\sigma_p \in A} \text{PSNR}(\hat{x}_\phi(\sigma_p), x) \tag{6}$$

using a Gaussian process (GP) as surrogate $f \sim \mathcal{GP}$. It is also possible to directly optimize the shape parameters of the prior. In each step of the BO, we evaluate our objective function $f$ at the current candidate $\sigma_p^*$ to increase the set of observations $\mathcal{D}_{\text{BO}}$ and update the posterior of the surrogate model. Next, we maximize an acquisition function $a(\sigma_p; \mu_{\mathcal{GP}}, \sigma_{\mathcal{GP}}^2)$ using the current GP posterior mean $\mu_{\mathcal{GP}}$ and variance $\sigma_{\mathcal{GP}}^2$. Its maximizing argument $\sigma_p^* \leftarrow \arg\max\ a(\sigma_p; \mu_{\mathcal{GP}}, \sigma_{\mathcal{GP}}^2)$ is used as candidate for the next iteration (Frazier, 2018). We choose the commonly accepted expected improvement (EI) as acquisition function

$$a_{\text{EI}}(\sigma_p; \mu_{\mathcal{GP}}, \sigma_{\mathcal{GP}}^2)) = \mathbb{E}\left[\max(y - f^*), 0) \,|\, y \sim \mathcal{N}(\mu_{\mathcal{GP}}(\sigma_p), \sigma_{\mathcal{GP}}^2(\sigma_p))\right] , \tag{7}$$

where $f^* = f(\sigma_{p,\text{best}})$ is the minimal value of the objective function observed so far. Eq. (7) can be solved analytically as shown in (Jones et al., 1998). We utilize automatic differentiation from modern deep learning frameworks to optimize the acquisition function in order to get the next candidate $\sigma_p^*$ (Gardner et al., 2018).

## 3. Experiments

We evaluate the performance of our MFVI approach on the following three inverse post-processing tasks and compare it to non-Bayesian DIP (Lempitsky et al., 2018), DIP with SGLD (Cheng et al., 2019) and DIP with MC dropout (Laves et al., 2020b). We apply BO to optimize the variance $\sigma_p$ of a Gaussian prior per task. In the following experiments, we use the same network architectures as proposed by (Lempitsky et al., 2018).

**Denoising**  Optical coherence tomography and ultrasound are prone to speckle noise due to interference phenomena, which can obscure small anatomical details and reduce image contrast. Speckle noise can be modeled as additive white Gaussian noise on log-transformed image intensities (Michailovich and Tannenbaum, 2006). Noise in low-dose X-ray originates from irregular photon density and can be modeled with Poisson noise (Lee et al., 2018; Žabić et al., 2013). We approximate the Poisson noise with Gaussian noise since $\text{Poisson}(\lambda)$ approaches a Normal distribution as $\lambda \rightarrow \infty$. We first create a low-noise image $x$ by smoothing and downsampling the original image to $256 \times 256$ pixel. This averages over highly correlated neighboring pixels affected by uncorrelated noise and decreases the observation noise. The downsampled image acts as ground truth and is corrupted by $\tilde{x} = x + \mathcal{N}(0, 0.1^2 I)$ using normal (X-ray) or log-transformed intensities (US and OCT). We use retinal OCT scans and chest X-rays with native resolutions of $496 \times 496$ and $1029 \times 1260$ pixel from a public data set (Kermany et al., 2018).

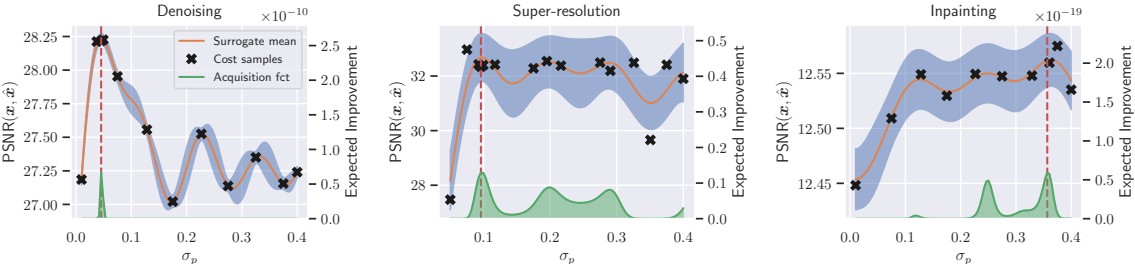

Figure 2: Results of Bayesian optimization. The acquisition function selects the next candidate for $\sigma_p$ based on the maximum of the expected improvement.

**Super-Resolution** In CT and MRI, the sampling frequency is limited due to inherent physical limitations of the imaging utility, i.e. the pitch or spacing of the detector (Greenspan, 2009). The resolution can be enhanced by reducing the size of detectors, but this comes at the expense of increased noise. Since imaging devices are usually tuned towards low noise and short acquisition time, part of the resolution is sacrificed. This motivates resolution-enhancing post-processing methods using a single image. We use slices of T1-weighted in vivo whole brain MRI with isotropic resolution of 250 μm (Lüsebrink et al., 2018) from public data sets. The $512 \times 448$ pixel full-resolution images act as ground truth $\boldsymbol{x}$ and are downsampled by a factor of 4 to obtain low-resolution images $\tilde{\boldsymbol{x}}$. The DIP network is optimized by applying a downsampling operator $d : \mathbb{R}^{4H \times 4W} \rightarrow \mathbb{R}^{H \times W}$ to its output $\hat{\boldsymbol{x}}$ and plugging $d(\hat{\boldsymbol{x}})$ into Eq. (4). To use gradient-based optimization, the downsampling operator must be differentiable and we opt for a Lanczos kernel (Duchon, 1979).

**Inpainting** Applications of inpainting in medical imaging are hair removal in dermoscopy (Abbas et al., 2011), specular highlight removal in endoscopy (Arnold et al., 2010), or metal artifact removal in CT sinograms (Peng et al., 2020) and MRI (Armanious et al., 2020). In this paper, we focus on the former task and sample images from the HAM10000 data set (Tschandl et al., 2018) showing different skin lesions with hair occlusions. We manually mask the hair and optimize the ELBO with zero-weighting the masked pixels in the likelihood term. The networks thus interpolate the masked areas.

### 3.1. Results

The results are presented as follows: First, we use BO to optimize the weight prior standard deviation $\sigma_p$ per task domain and use the optimal value in the subsequent experiments. Next, we show that all competing methods overfit the low-quality image given enough iterations. Our method outperforms the other methods by means of reconstruction accuracy on all tasks and modalities after convergence when using an optimized weight prior and provides well-calibrated predictive uncertainty maps.

Fig. 2 shows that the optimal $\sigma_p$ for denoising and super-resolution imposes a narrow prior with $\sigma_{p,\text{den}}^* = 0.05$ and $\sigma_{p,\text{sr}}^* = 0.1$, respectively. In inpainting, the optimal value $\sigma_{p,\text{inp}}^* = 0.36$ is slightly higher. A narrow prior prevents weights from growing large, effectively avoiding overfitting of the corrupted image (note that we fixed $\mu_p = 0$).

This is empirically shown in Fig. 3, where DIP and DIP with SGLD strongly overfit the corrupted patterns, making manually applied early stopping essential to obtain the highest reconstruction accuracy (indicated by the narrow peaks). Additionally, MC dropout overfits at some point, although the peak is wider and overfitting starts later in optimization. While the PSNR between reconstruction $\hat{x}$ and ground truth $x$ approaches the PSNR between noisy image $\tilde{x}$ and ground truth, for DIP and SGLD, MFVI safely converges to the optimal value in all modalities. In super-resolution, overfitting is less severe. MCDIP and MFVI do not overfit the low resolution image. DIP and SGLD do not show a sharp peak but rather decline slowly as shown in Fig. 3 (right) and Fig. 5. MFVI consistently provides well-calibrated pixel-wise uncertainty in denoising and super-resolution (see Tab. 1 in appendix).

For inpainting we restrict ourselves to a qualitative view onto the reconstruction results as all approaches converge to similar PSNR in the non-masked regions. While the reconstruction of the DIP and SGLD contain artifacts, MC dropout and MFVI produce very smooth reconstruction results. In inpainting tasks, the uncertainty maps are especially interesting, which we expect to show high uncertainty in masked regions as the model does not receive information from these areas. It can be seen in Fig. 4 that the reconstruction of MFVI exhibit high uncertainty in regions with hair, while showing lower uncertainty in the higher frequency regions of the chloasma. In the region of the lesion, the uncertainty should be as low as possible, as it is important for the downstream task of classifying the skin lesion.

## 4. Conclusion

We presented a mean-field variational inference approach to deep image prior and optimized the weight prior using Bayesian optimization. Bayesian methods are in general more robust to overfitting due to their inbuilt regularization from the weight prior. However, a badly selected prior can still cause overfitting (as shown empirically for SGLD and MC dropout). MFVI allows for a more detailed prior selection, which we exploit to optimize the prior using Bayesian optimization. Selecting a suitable prior fixes the overfitting behavior of DIP-based approaches, which are generally interesting for medical imaging, as no supervised training is required. Different inverse post-processing tasks in medical imaging were performed to show the benefits of the proposed method. BO was used to optimize the prior towards reconstruction accuracy. Even if early stopping is applied to the other methods, our approach performs on-par with respect to reconstruction accuracy and yields well-calibrated uncertainties. It is further possible to additionally optimize the prior with respect to a calibration metric to ensure well-calibrated uncertainty maps. The presented approach is not limited to post-processing tasks and can also be used for CT or MRI reconstruction from sinograms.

## Acknowledgments

MT is supported by Informatics for Life founded by the Klaus Tschira Foundation, ML and AS received funding from Interdisciplinary Competence Center for Interface Research (ICCIR).

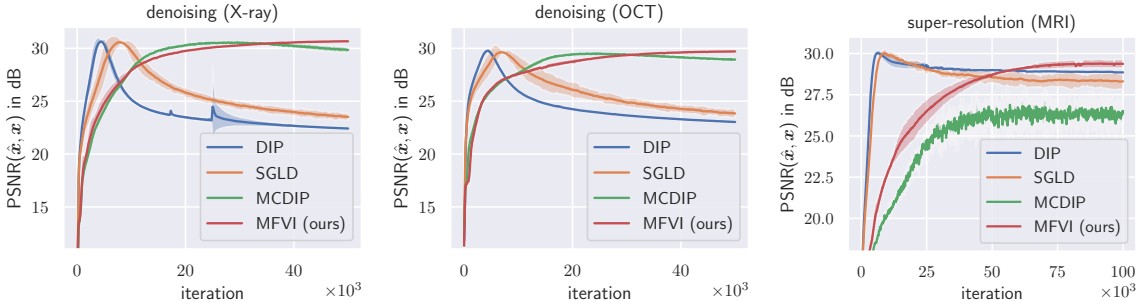

Figure 3: Our MFVI approach with an optimized prior does not overfit. Plots show mean $\pm 2\times$ standard deviation from 3 runs with different random initialization.

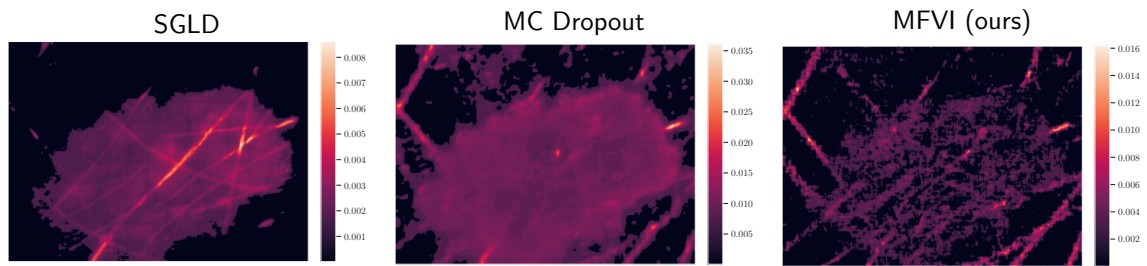

Figure 4: MFVI shows high uncertainty in masked regions for inpainting, whereas the other methods show high uncertainty in regions important for the downstream task.

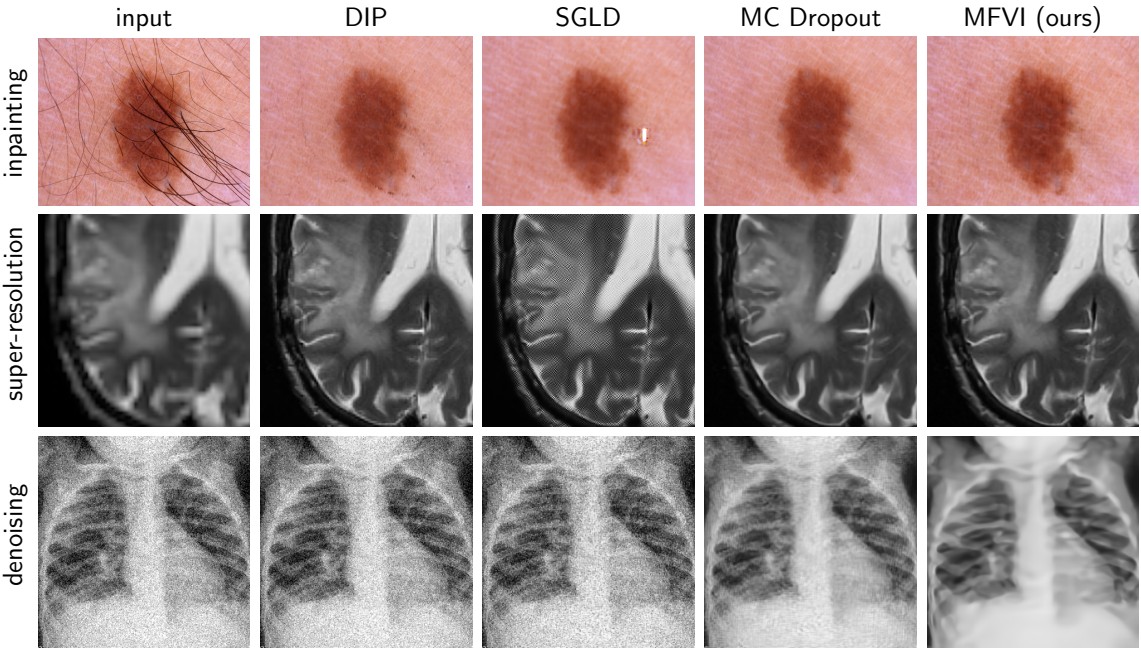

Figure 5: Qualitative results for the different tasks after convergence.

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

## Appendix A. KL Divergence Between Two Gaussians

If a Gaussian prior is chosen for convenience, the KL divergence is analytically tractable (cf. Eq. (3)). Let $p(x) = \mathcal{N}(\mu_p, \sigma_p^2)$ and $q(x) = \mathcal{N}(\mu_q, \sigma_q^2)$. It is known that

$$
\begin{aligned}
\mathrm{KL}[q(x) \,\|\, p(x)] &= \int q(x) \log \frac{q(x)}{p(x)} \,\mathrm{d}x = \int q(x) \log q(x) \,\mathrm{d}x - \int q(x) \log p(x) \,\mathrm{d}x \\
&= -\frac{1}{2}\left(1 + \log 2\pi\sigma_q^2\right) + \frac{1}{2} \log 2\pi\sigma_p^2 + \frac{\sigma_q^2 + (\mu_q - \mu_p)^2}{2\sigma_p^2} \\
&= \log \frac{\sigma_p}{\sigma_q} + \frac{\sigma_q^2 + (\mu_q - \mu_p)^2}{2\sigma_p^2} - \frac{1}{2} \ .
\end{aligned}
$$

## Appendix B. Computational Complexity of MFVI

Prior selection using Bayesian optimization is performed offline and does not have to be repeated for each image at hand. Therefore, increased complexity can be attributed to the parameter sampling of MFVI. In each forward pass, the additional steps are (1) drawing $n$ samples from a univariate Gaussian, where $n$ is the number of parameters of the convolutional autoencoder and (2) reparameterization of the actual parameters by $w_i = \mu_i + \sigma_i \epsilon_i$, which results in $n$ additional multiplications and additions. In our experiments, this results in $\approx 2\times$ slower forward pass times and $2\times$ increased memory footprint. Relative wall times for the denoising task were 1.0 for non-Bayesian DIP, 1.13 for MC dropout, 2.20 for MFVI and 2.76 for SGLD.

## Appendix C. Additional Figures

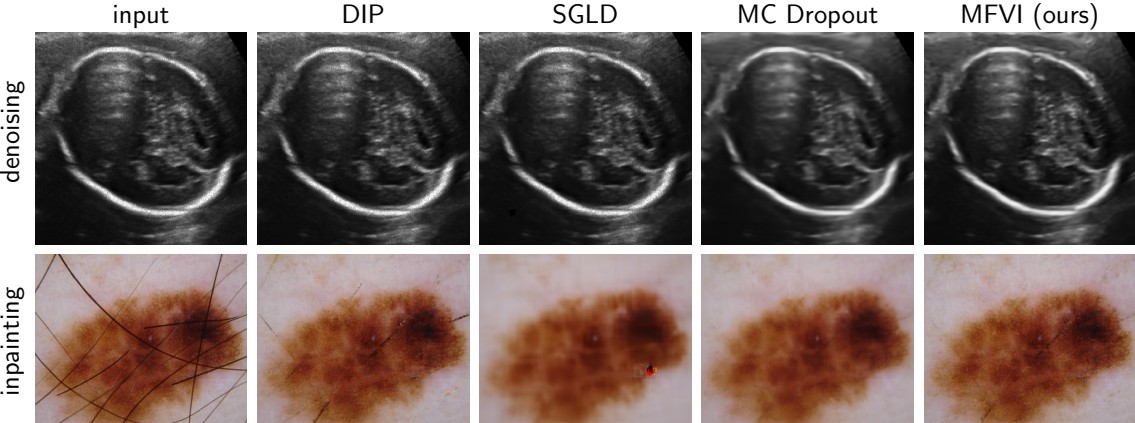

Figure 6: Additional qualitative results for US denoising and hair inpainting after convergence. The reconstructions from MC dropout and MFVI look most valid, while MC dropout overly smoothes important details (cf. texture of skin lesion).

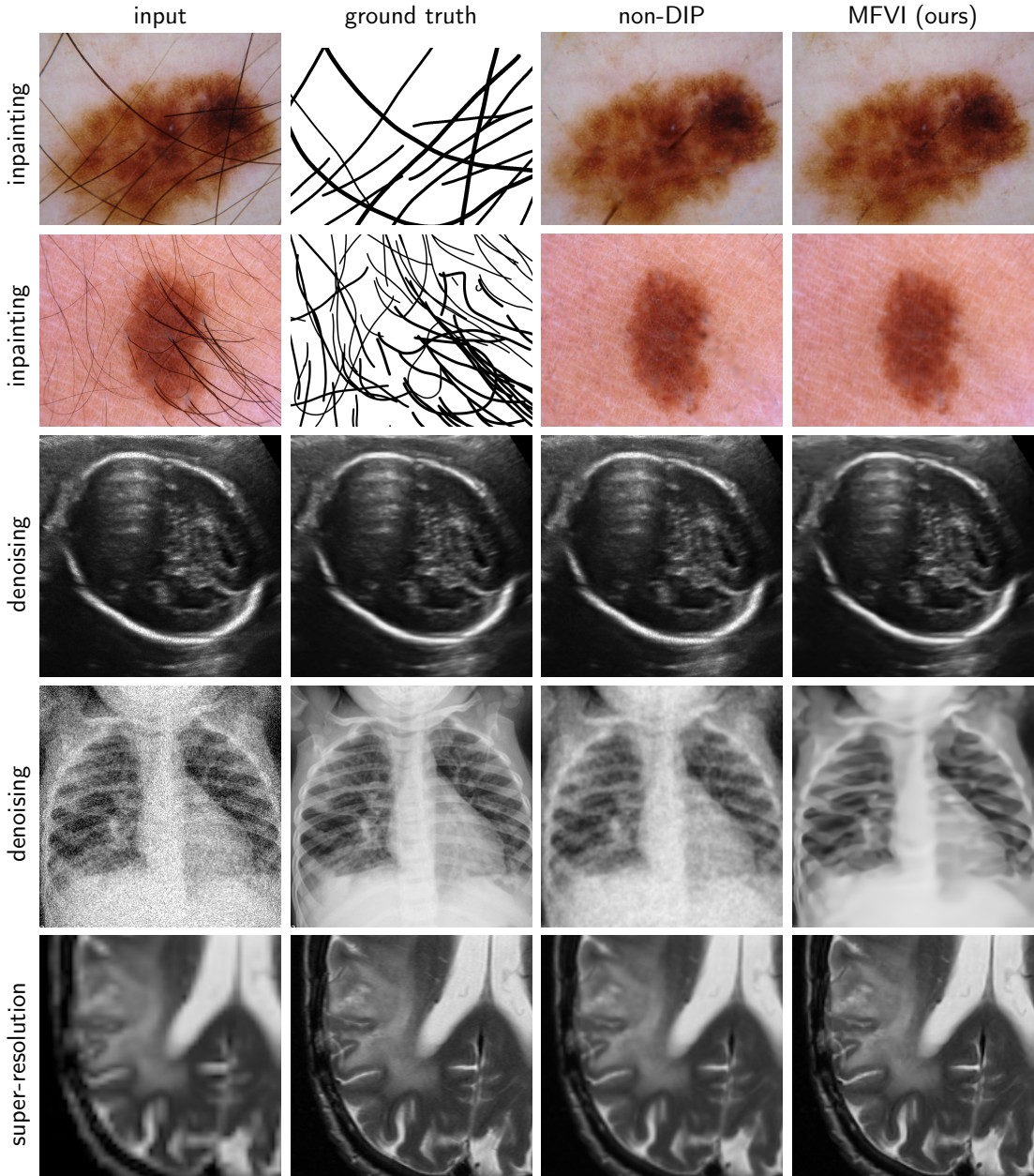

Figure 7: Non-DIP method comparison. The non-DIP algorithms are biharmonic functions for inpainting (Damelin and Hoang, 2018), anisotropic diffusion for denoising (Perona and Malik, 1990) and bilinear interpolation for super-resolution.

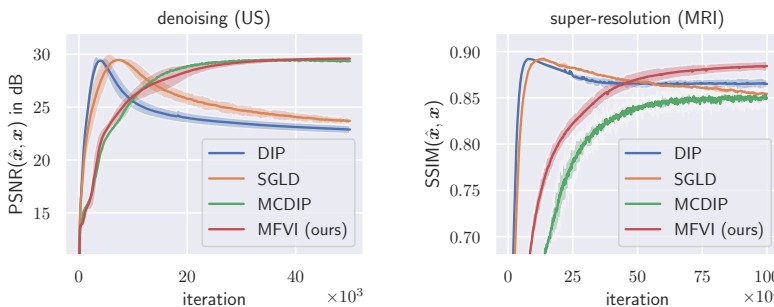

Figure 8: (Left) Additional PSNR curve for denoising on US. (Right) Results for super-resolution measured using structural similarity index measure (SSIM).

## Appendix D. Uncertainty Calibration

Table 1: Uncertainty calibration error (UCE) (Laves et al., 2020a) for denoising and super-resolution experiments. The UCE describes the expected discrepancy between pixel-wise error and uncertainty of the reconstructions.

|  | SGLD | MCDIP | MFVI (ours) |
|---|---|---|---|
| denoising (X-ray) | 0.915 | 0.258 | **0.093** |
| denoising (OCT) | 0.815 | 0.144 | **0.073** |
| denoising (US) | 0.799 | 0.309 | **0.134** |
| super-res. (MRI) | **0.012** | 0.349 | 0.069 |

## Appendix E. Illustration of Mathematical Concept

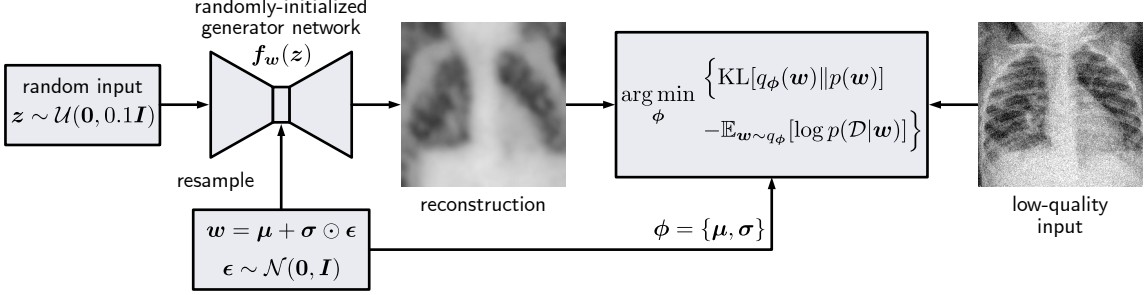

Figure 9: Illustration of the mathematical concept behind MFVI DIP.

**Pseudocode of MFVI DIP**

1. Sample input $\boldsymbol{z}' = \mathcal{U}(0, 0.1)$

2. While $i < i_{\max}$ do

   (a) Permute input $\boldsymbol{z} = \boldsymbol{z}' + \mathcal{N}(0, 0.01)$

   (b) Sample $\boldsymbol{\epsilon} \sim \mathcal{N}(\mathbf{0}, \boldsymbol{I})$

   (c) Let $\boldsymbol{w} = \boldsymbol{\mu} + \boldsymbol{\sigma} \odot \boldsymbol{\epsilon}$ with variational parameters $\boldsymbol{\phi} = \{\boldsymbol{\mu}, \boldsymbol{\sigma}\}$

   (d) Compute loss $\mathrm{ELBO}\left(\boldsymbol{f_w}(\boldsymbol{z})\right) = \log q_{\boldsymbol{\phi}}(\boldsymbol{w}) - \log p(\boldsymbol{w}) - \log p(\mathcal{D} \,|\, \boldsymbol{w})$

   (e) Compute the gradient w.r.t. the mean and standard deviation

   $$\Delta_{\boldsymbol{\mu}} = \frac{\partial \mathrm{ELBO}\left(\boldsymbol{f_w}(\boldsymbol{z})\right)}{\partial \boldsymbol{w}} + \frac{\partial \mathrm{ELBO}\left(\boldsymbol{f_w}(\boldsymbol{z})\right)}{\partial \boldsymbol{\mu}}$$
   $$\Delta_{\boldsymbol{\sigma}} = \frac{\partial \mathrm{ELBO}\left(\boldsymbol{f_w}(\boldsymbol{z})\right)}{\partial \boldsymbol{w}} + \frac{\partial \mathrm{ELBO}\left(\boldsymbol{f_w}(\boldsymbol{z})\right)}{\partial \boldsymbol{\sigma}}$$

   (f) Update the variational parameters $\boldsymbol{\phi}$

   $$\boldsymbol{\mu} \leftarrow \boldsymbol{\mu} - \eta \Delta_{\boldsymbol{\mu}}$$
   $$\boldsymbol{\sigma} \leftarrow \boldsymbol{\sigma} - \eta \Delta_{\boldsymbol{\sigma}}$$

   (g) $i \leftarrow i + 1$

## Appendix F. Background on Bayesian Deep Learning

In Bayesian deep learning, a prior distribution $p(\boldsymbol{w} \,|\, \alpha)$ is placed over the weights $\boldsymbol{w}$ of a neural network, governed by a hyperparameter $\alpha$. After observing the data $\mathcal{D}$, we are interested in the posterior $p(\boldsymbol{w} \,|\, \mathcal{D}, \alpha) = p(\mathcal{D} \,|\, \boldsymbol{w}, \alpha) p(\boldsymbol{w} \,|\, \alpha) / p(\mathcal{D})$. However, this distribution is not tractable in general. This gives rise to different approximate Bayesian inference techniques that rely on either sampling or VI. SGLD is a framework that derives a Markov chain Monte Carlo (MCMC) sampler from SGD by injecting Gaussian noise into the gradients after each learning step (Welling and Teh, 2011). Under suitable conditions SGLD eventually converges to the posterior distribution. VI uses optimization instead of sampling to find the member $q_{\boldsymbol{\phi}}(\boldsymbol{w})$ of a family of distributions (e.g. a multivariate Gaussian) that is close to the exact posterior, defined by the variational parameters $\boldsymbol{\phi}$. We optimize $q_{\boldsymbol{\phi}}$ w.r.t. $\boldsymbol{\phi}$, such that the Kullback-Leibler divergence is minimized with regard to the true posterior (Blei et al., 2017). Two practical implementations are MC dropout (Gal and Ghahramani, 2015) and Bayes by backprop (Blundell et al., 2015). The former uses dropout before every weight layer during training and at inference time, which allows sampling from the approximate posterior. The latter assumes a fully factorized Gaussian distribution $w_{ij} \sim \mathcal{N}(\mu_{ij}, \sigma_{ij}^2)$, also known as mean-field distribution, which treats the mean and variance of each weight as learnable parameter. In contrast to SGLD and MC dropout, MFVI allows us to directly compute the KL divergence between the variational posterior and the prior, which enables us to select other (non-Gaussian) prior distributions, where no closed form exists.

