# OpenReview forum: "A Mean-Field Variational Inference Approach to Deep Image Prior for Inverse Problems in Medical Imaging"
_MIDL.io/2021/Conference — MIDL 2021_

### Official Review · AnonReviewer3 · 2021-02-25

**Confidence:** 3
**Preliminary Rating:** 3
**Recommendation:** Poster
**Final Rating:** 3

**Summary:**

In this paper the authors propose a combination of Mean-field Variational inference and the concept of deep image priors, applying it to the medical imaging field. Motivation of the paper boils down to avoiding hallucinations by CNNs on generative tasks.  They evaluate their method on denoising, super-resolution and inpainting tasks.

**Strengths:**

Motivation: The paper is relatively clear on what it tries to achieve and via what method. The motivation of the chosen method is clear as well.

Structure & Language : The language used in the paper is proper and without any major issues. The structure is ok albeit not great. See weaknesses for more on this point.

Related work: Adequate but not great prior work analysis, missing were specifically works relating to non bayesian approaches to the tasks at hand and how they fair in comparison to your work.

Proposed Method : MFVI and DIP are two well known and powerful techniques. The authors proposed a combination of the two following the path of combining DIP with Bayesian approaches set out by prior work (Laves et al. 2020b), (Cheng et al., 2019);

Experimentation: The technique appears to work relatively well, although figs 2,3 are quite small and hard to read properly.



**Weaknesses:**

Structure & Language: As i said earlier there could be a better distinction between introduction/motivation and the related works

Contributions: Taking the claims made by the authors 1-1.
a) "The paper shows that former bayesian DIP approaches overfit on medical images". Technically speaking this is only showed through a single figure in the paper, empirically derived. In addition overfitting of DIP with SGLD is shown by Laves et al. (2020b) as correctly pointed out by the authors. Hence the resulting contribution is that DIP with MCDropout overfits. " We attribute this to the manual selection of the weight’s prior distribution." it would be to the benefit of the paper if this discussion was more explicit and to the point. For example the "contributions" paragraph is not the proper place to discuss the shortcomings of other methods. The supporting evidence for this claim are sparse and mixed in the greater paper making it difficult to follow.

b) the second claim regarding the combination of MFVI and DIP has adequate supporting evidence

Experimentation: Experimentation and comparison with at least 1 non bayesian non DIP method on each task would be a benefit. That would help drive the point on why the need for the MFVI approach.

**Deanonymize Review:**

no

**Final Rating Justification:**

I thank the authors for their rebuttal and the changes they made to the manuscript. I maintain my acceptance rating.

**Justification Of The Preliminary Rating:**

Overall: this is an ok paper, areas of improvement can be found above. It offers a natural progression of prior work, without any major breakthroughs or surprising results. Structure and comparisons could have been thought out a bit better. But most importantly the structure of an argument is not followed in one of the two claims made in the paper.

**Paper Type:**

methodological development

**Special Issue:**

no

---

> ### Author Response · Authors · 2021-03-15
> **Restructure First Section and Detailed Figures**
>
> Dear Reviewer3, thank you very much for your useful feedback! We will revise our paper with regard to the aspects you mentioned and hope to have clarified all open questions.
>
> 1. We will restructure the first section by including a paragraph about related work.
> 2. Due to space limitations, we did not include additional plots showing the overfitting behavior of DIP with SGLD and MC Dropout. We will add more figures to the appendix showing that the methods do overfit on other modalities as well. Also, all plots in Fig. 3 show that DIP with SGLD performs worse than DIP with MFVI. DIP with MC Dropout is only comparable in the domain of super-resolution, where we expect to see overfitting with more iterations as well, since standard DIP does also only overfit much slower in super-resolution than in denoising.
> 3. As already answered to Reviewer1: Bayesian methods are generally more robust to overfitting due to their inbuilt regularization from the weight prior. As shown empirically a handcrafted badly selected prior can still cause overfitting for DIP with SGLD and MC dropout. MFVI-DIP allows for a more detailed prior selection because of the explicit calculation of the KL divergence, which we exploit to optimize the prior using Bayesian optimization. We will add this to our discussion in the final manuscript.
> 4. We will move the discussion of shortcomings from other methods to the related work paragraph.
> 5. To compare our proposed methods with non-Bayesian non-DIP methods we incorporate anisotropic diffusion for denoising [1], bilinear-upsampling for super-resolution and inpainting by biharmonic functions [2].
>
> References
>
> [1] Perona, P., Malik, J. (1990). Scale-space and edge detection using anisotropic diffusion. IEEE TPAMI, 12(7), 629-639. http://image.diku.dk/imagecanon/material/PeronaMalik1990.pdf
>
> [2] Damelin, S. B., Hoang, N. S. (2018). On surface completion and image inpainting by biharmonic functions: Numerical aspects. International Journal of Mathematics and Mathematical Sciences. https://arxiv.org/abs/1707.06567

---

### Official Review · AnonReviewer4 · 2021-03-06

**Confidence:** 4
**Preliminary Rating:** 3
**Recommendation:** Poster

**Summary:**

In this paper, the authors propose Mean-Field Variational Inference (MFVI) approach for addressing the problem of manual early stopping at the right stage in the optimization process in Deep Image Priors (DIP) in order to not overfit image details.  By having a well-defined prior distribution in a Bayesian setting, i.e MFVI and Bayesian optimization (BO) to tune the parameters of the weight prior, the quality of the reconstructed images in the inversion problems can be improved.
The optimization objective for the MFVI approach has two minimization terms - 1. The maximum posterior estimate terms that minimizes the negative log likelihood.This term gives out the reconstructed output together with the pixel-wise uncertainty. 2. The optimal variational parameters (the mean and the variance) of the model weights by minimizing the negative log evidence lower bound (ELBO) i.e the KL divergence between the variational posterior and the prior.
For choosing an appropriate prior distribution over the model weights, the proposed method uses Bayesian optimization with Gaussian Process (GP) as the surrogate for maximizing the PSNR between the reconstruction output and the target.


**Strengths:**

Convolutional neural networks with supervised learning exhibit hallucinations even when there is a small shift in the data distribution at test time. The proposed algorithm addresses the important problem of over-fitting that arises from data driven approaches.

The proposed method is based on the concept of deep image priors, in this case an untrained image-generator network, and outputs both the reconstructed image and the uncertainty estimates of the weights of the network given the data.

A separate experiment for the Bayesian optimization for the choice of the prior as against the manual early stopping in the compared methods helps to understand the contribution of the work better.

The Bayesian aspect is emphasized well with good theoretical background explanations.

The motivation of the problem in the introduction part is explained well.


**Weaknesses:**

The authors mention in the introduction section that, in Lempitsky et al. (2018), the structure of the network also decides the ability of the network to learn image priors from data. This would bias the reader to expect experimental analysis on the structure of the network on the performance. The structure of the DIP is not mentioned in the experiment section at all. Also the experiments focus only on the application of inverse imaging tasks and do not reflect the effect of the structure of the DIP considered. Is the image generator network a convolutional neural network?

In the non-Bayesian DIP, the size of the training set is generally mentioned. This is because the performance of the learning based DIP depends on the number of training images.

Visual quality comparison in Figure 5 should highlight the metric values, since visually the images for DIP and MFVI show only minor variations in the case of super-resolution. Also the ground truth images are not shown. Similarly standard quantitative metrics used for in-painting must be provided.

The authors should add a pseudo code in the Appendix section to understand the overall flow of the posterior weight estimation to add clarity for interested readers. This holds also for the Bayesian optimization procedure in Section 2.2

While the proposed method shows visually pleasing results for in-painting and denoising (removal of unwanted details), for inverse tasks like super-resolution, the recovery (or preservation) of details is very important. This aspect would be reflected better with the structure similarity (SSIM) metric with respect to the target. It is not clear why the authors have not taken this aspect into consideration.

In Section 2.1, “ In each forward pass, the weights are sampled using reparameterization.” --- it is not clear if the same image or different image is used in each forward pass.

In Section 3.1 - Our method outperforms the other methods by means of reconstruction accuracy on all tasks and modalities after convergence when using an optimized weight prior and provides well-calibrated predictive uncertainty maps. However the authors have not shown the use of any calibration metric to ensure well-calibrated uncertainty maps.


**Deanonymize Review:**

no

**Detailed Comments:**

There are a few corrections.

In Fig.2 the multiplication factors for denoising and inpainting are specified at the peaks. For super-resolution this factor is missing.

In Acquisition fct is not clear. This must be corrected.




**Justification Of The Preliminary Rating:**

This work addresses the important problem of over-fitting inherent in data driven approaches using convolutional neural networks (CNN). The problem of hallucinations  - both false positives and false negatives in supervised learning is taken into account.  The concept of DIP takes into account an untrained network as the image generator and uses the generator to derive image priors using Bayesian approach. The design of experiments is really good and explains the actual contribution of this work.

**Paper Type:**

methodological development

**Questions To Address In The Rebuttal:**

It is not stated clearly how the assumption that the variational posterior can be factorized based on the number of layers. Does this mean that the layer weight densities are independent?

The following sentence in the contributions is unclear.
It is important to distinguish between DIP, which imposes a spectral bias towards lower frequencies, and the prior distribution over the weights of the network in Bayesian inference.
Do the authors mean that the distribution of weights in an Auto-encoder like DIPs is not the same as the prior distribution of the weights used in a Bayesian inference approach?

In the background the use of notation $q_{phi}(theta)$ is unclear. It must be $q_{phi}(w)$?

Are different DIPs used for different restoration tasks?

For denoising in Fig 3, the PSNR for the proposed method crosses over that of the reconstructions of MCDIP for denoising (X-ray) and super-resolution (MRI), this is not the case for denoising (OCT). Does this need more iterations to expect the same behavior as the other two cases?

In the Denoising part of the Experiments section, “ We first create a low-noise image x by smoothing and downsampling the original image to 256×256 pixel.” ----> What is the original image size?




**Special Issue:**

no

---

> ### Author Response · Authors · 2021-03-16
> **Response**
>
> Dear Reviewer4, your thorough review allows us to further improve our work. Below, we respond to each of your comments and hope to address all concerns and questions. We will incorporate your suggested changes into an updated version of our manuscript.
>
> 1. We use the identical structure as Lempitsky et al. (2018), which was also used by Cheng et al. (2019) and Laves et al. (2020), to ensure comparability between the approaches. The DIP network consists of convolutional, upsampling and batch normalization operations, hence, it is a convolutional generator network. We will include a sentence in our manuscript for clarification.
> 2. The main advantage of DIP is its "training" on one example only; we do not train on a data set in either the Bayesian or non-Bayesian setting. Therefore, the performance of all DIP approaches does not depend on a number of training images. It can be interpreted as non-learning method using convolutional networks.
> 3. We will add the ground truth versions and quantitative metrics to the images shown in Fig. 5, especially in the case of super-resolution. We do not use a quantitative metric for inpainting as there is no single objective measure to quantify inpainting results [1]. Further, we do not have a ground truth for inpainting, which makes qualitative inspection mandatory.
> 4. We will add pseudocode and a graphical illustration (see question from Reviewer2) for clarification of Bayesian optimization and the posterior weight estimation to the appendix, which explains the sampling of weights and usage of ELBO in the Bayesian setting.
> 5. We did not show the figures for SSIM as the results are very similar to the PSNR curves, but we will add them to the updated version of our paper.
> 6. The input to the DIP network is uniformly distributed noise that is fixed ($ \mathbf{z} \sim \mathcal{U}(0, 0.1) $). However, the distribution of weights is not fixed and optimized during training. Each weight is sampled with the local reparameterization trick in every forward pass with a fixed input $ \mathbf{z} $. The corrupted (noisy, low-resolution, masked) image $ \tilde{\mathbf{x}} $—to which the loss is computed—stays the same during all iterations as well.
> 7. We assessed the validity of the uncertainty maps only qualitatively. To address your comment, we will compute and provide an appropriate calibration metric (e.g., ECE, UCE) for all Bayesian approaches.
> 8. We do also use the maximum of the surrogate model of the Bayesian optimization in Fig. 2. This is just not as unambiguously as the ones for denoising and inpainting.
> 9. The heart of mean-field variational inference is the assumption of factorized weights (layers). By discarding covariances, each weight is independent and not influenced by the variance of others. The lower computation time comes at the expense of not exact variance estimation. However, it is usually enough to capture the mode of the posterior, where VI is mostly sufficient. See [2] for additional information about MFVI.
> 10. There is no distinction between the distribution of weights used in an autoencoder like DIP and the posterior distribution in Bayesian inference, which is regularized by the prior. Because of its architecture, DIP acts as a low pass filter. However, this low pass behavior diminishes during training of standard DIP as higher frequencies are also reconstructed in later training stages, leading to overfitting by also reconstructing the noise. The distributions over weights in Bayesian inference preserves the low pass behavior by mitigating overfitting.
> 11. Thank you for pointing out the typo. $ q_{\phi}(w) $ is indeed correct.
> 12. We follow the protocol proposed by Lempitsky et al. (2018). The architectures used are slightly different (e.g., with or without skip connections) but a convolutional encoder-decoder hourglass like structure is always used.
> 13. We expect to see the same behavior for OCT as for X-ray. In our updated version we will include the corresponding PSNR curve.
> 14. The original size was twice as big; we will also include that point.
>
> We thank you again for your detailed comments and are working diligently on an updated version of our manuscript that includes all changes.
>
> References
>
> [1] Oncu, A. I., Deger, F., Hardeberg, J. Y. (2012). Evaluation of Digital Inpainting Quality in the Context of Artwork Restoration. ECCV, 561-570. https://doi.org/10.1007/978-3-642-33863-2_58
> [2] Blei, D. M., Kucukelbir, A., & McAuliffe, J. D. (2017). Variational inference: A review for statisticians. Journal of the American Statistical Association, 112(518), 859-877. https://doi.org/10.1080/01621459.2017.1285773

---

### Official Review · AnonReviewer2 · 2021-03-06

**Confidence:** 2
**Preliminary Rating:** 3
**Recommendation:** Oral, Poster
**Final Rating:** 4

**Summary:**

In this paper, the authors explore inverse imaging problems like denoising, super-resolution or inpainting in the context of medical imaging.
For those problems, deep convolutional networks have shown promising performance. However, they suffer from a spectral bias towards low frequencies, which require interventions like early stopping to avoid problems with high frequencies. To address this, the authors propose a mean-field variational approach to learn the weight priors and validate this approach on the three tasks denoising, super-resolution, and inpainting. Results obtained are similar or better than previous works but the training requires less human intervention as early-stopping is not necessary anymore.

**Strengths:**

The paper is well written and structured. It was easy and enjoyable to read.

The research problem is well introduced and the proposed work well positioned against previous articles.
There was a real effort in guiding the reader and explaining the method clearly. Even though I am not really familiar with Bayesian Optimisation, I could understand the proposed method.
The experimental setup is well designed and the results interesting.


**Weaknesses:**

I am missing a word on the complexity of the proposed approach VS the previous works. The proposed approach seems to require more steps and calculations than previous works but it is compensated by less human intervention required. It would be good to discuss this point somewhere.

Minor comments:
- It would be good to add more details on the image sizes to see if this approach is suitable for high-resolution images.
- A figure with a graphical illustration of the mathematical concepts would help the reader to understand the idea.
- It would be good to have more qualitative results in the appendix to compare the output of the proposed methods to previous works.

**Deanonymize Review:**

no

**Detailed Comments:**

Nothing to add to my previous feedback.

**Final Rating Justification:**

The authors did a good work in the rebuttal. They answered all my questions/concerns and provided an updated version of the article. For this reason, I'm increasing my rating to Strong Accept.

**Justification Of The Preliminary Rating:**

The paper is overall of very good quality, it is well structured and written. The form could be improved to make the paper easier to read and the complexity a bit more discussed. However, this can be easily done and I consider the paper already ready for acceptance.

**Paper Type:**

methodological development

**Questions To Address In The Rebuttal:**

Any answer to my previous feedback would be helpful.

**Special Issue:**

yes

---

> ### Author Response · Authors · 2021-03-16
> **About computational complexity**
>
> Dear Reviewer2, we appreciate your detailed and helpful review. We will address each of your issues in the following and hope to answer all open questions.
>
> 1. Thank you for pointing out missing discussion about the complexity of our approach. We will add discussion about the computational complexity to our manuscript. Prior selection using Bayesian optimization is performed offline and does not have to be repeated for each image at hand. Therefore, increased complexity can be attributed to the parameter sampling of MFVI. In each forward pass, the additional steps are (1) drawing $ n $ samples from a univariate Gaussian, where $ n $ is the number of parameters of the convolutional autoencoder and (2) reparameterization of the actual parameters by $ w_{i} = \mu_{i} + \sigma_{i} \epsilon_{i} $, which results in $ n $ additional multiplications and additions. In our experiments, this results in $ \approx 2 \times $ slower forward pass times compared to non-Bayesian DIP.
> 2. We will add more information about the image sizes, which was also pointed out by other reviewers.
> 3. A graphical illustration of the mathematical concepts will be added to the appendix along with a pseudocode summary (see our response 4 to Reviewer4).
> 4. We will add more qualitative results for additional images in the appendix. As suggested by Reviewer3, we will also add qualitative results for non-Bayesian non-DIP methods.

---

### Official Review · AnonReviewer1 · 2021-03-08

**Confidence:** 3
**Preliminary Rating:** 3
**Recommendation:** Oral, Poster
**Final Rating:** 3

**Summary:**

This work proposes a Bayesian approach to deep image priors that uses Bayesian optimisation to learn the weight priors. With this, the authors claim that early stopping is not required and that any further training does not lead to increased overfitting.

This is tested on three inverse problem experiments: denoising, super-resolution and inpainting. The results show that further training iterations do not lead to overfitting.

**Strengths:**

The work is clearly motivated: inverse problems (e.g. inpainting and denoising) are common in medical imaging pipelines. Improving performance on this task is of clinical significance. The paper is well written and easy to follow.

The experiments demonstrate the claims set out by the authors.

**Weaknesses:**

Although there are no major weaknesses, the results could be presented with error bars from repeated runs. This would further increase confidence that the performance improvements are not due to chance (e.g. is the collapse SGLD on MRI data just an unlucky run? in figure 3).

My other issue is a claim in the contributions. The authors state that: "we show that all former Bayesian approaches to DIP show overfitting". However, it seems they test against two other Bayesian DIP methods. Many previous studies including [1] are not shown in the results.

[1] Carrillo, H., Millardet, M., Carlier, T., & Mateus, D. (2021, February). Low-count PET image reconstruction with Bayesian inference over a Deep Prior. In Medical Imaging 2021: Image Processing (Vol. 11596, p. 115960V). International Society for Optics and Photonics.

**Deanonymize Review:**

no

**Detailed Comments:**

- the x in the first set of equations in the introduction is not defined
- acronyms are defined but then the full length words are re-used later in the manuscript. For example variation inference is defined to be VI, yet "variational inference" is spelled out numerous times.
- More discussion on why this method overfits less would be helpful is space permits

**Final Rating Justification:**

The authors have responded to my questions and amended the manuscript to address my concerns. I have therefore upgraded my rating to a 3 (accept).

One additional comment: it is not clear what the blue shaded region in Figure 2 is. Please update the caption to clarify.

**Justification Of The Preliminary Rating:**

The paper is a valuable addition to the field. It is well written and clearly explains the problem and solution using bayesian optimisation for deep image priors. The applications to medical imaging are clear.

Revisions, as described above, can further improve this paper.

**Paper Type:**

methodological development

**Questions To Address In The Rebuttal:**

- Repeated runs to show that results are not due to chance, or a clear statement on that limitation in the discussion.
- Fixes to the claims set out in the contributions to match the results shown in the paper

**Special Issue:**

no

---

> ### Author Response · Authors · 2021-03-15
> **Repeated Runs**
>
> Dear Reviewer1, thank you very much for your valuable review. In the following, we will address each of your comments and hope to solve all open questions.
>
> 1. We are currently re-running our experiments multiple times and will provide results with mean and standard deviation (error bars) in an updated version of our manuscript.
> 2. Thank you for pointing out this recent and highly relevant prior work, which we have already read carefully and will consider in our paper. However, the authors use posterior inference with SGLD-DIP from Cheng et al. (2019), which we did include in our experiments. We will follow your advice and change our contribution claim accordingly.
> 3. Thank you for pointing out the use of x before its introduction. We will introduce x before its first occurence.
> 4. We will remove the use of the full length words after defining their acronyms.
> 5. Bayesian methods are in general more robust to overfitting due to their inbuilt regularization from the weight prior. However, a badly selected prior can still cause overfitting (as shown empirically for SGLD and MC dropout). MFVI-DIP allows for a more detailed prior selection, which we exploit to optimize the prior using Bayesian optimization. We will add this to our discussion in the final manuscript.

---

### Author Response · Authors · 2021-03-11
**General Response**

Dear reviewers, thank you very much for your comprehensive reviews and your kind verdict. Within the next week, we will address each review and comment individually and hope that we can clarify all open issues. If further questions arise, we welcome an open discussion.

---

> ### Author Response · Authors · 2021-03-18
> **Rebuttal version**
>
> Dear reviewers, we have just uploaded a rebuttal version of our manuscript, which already incorporates almost all comments, including repeated experiments, a discussion of computational complexity, pseudocode and a graphical illustration of the concept, a comparison with non-Bayesian non-DIP methods, and some minor changes. The remaining changes will be implemented in a possible camera-ready version.
>
> We thank you again for your valuable feedback, which helped us to considerably improve our paper. If we were able to address your concerns, we would greatly appreciate an updated rating of our submission.

---

### Meta-Review · Area_Chairs · 2021-03-25

**Recommendation:** Accept (Oral)

**Metareview:**

This paper introduces a new Bayesian approach to deep image prior using mean-field variational inference, where the goal is to enable pixel-wise uncertainty quantification, and mitigate the need for early stopping regularisation.  The paper defines a prior distribution in a Bayesian setting to learn the parameters of the weight prior, targeting an improvement in the quality of the reconstructed images for inversion problems (i.e., denoising, super-resolution and inpainting).  All reviewers demonstrated full support for the publication of the paper and the questions were more about clarifications about claims, experiments, etc.  I also support the publication of this paper, and encourage the authors to address the main comments by the reviewers.


**Paper Type:**

methodological development

---

### Decision · Program_Chairs · 2021-03-31

**Decision:**

Accept

**Comment:**

Congratulations your paper has been selected as a long oral.